# Scanning Electron Microscopy Investigation for Monitoring the Emulsion Deteriorative Process and Its Applications in Site-Directed Reaction with Paper Fabric

**DOI:** 10.3390/molecules26216471

**Published:** 2021-10-27

**Authors:** Liewei Qiu, Yongkang Zhang, Xueli Long, Zhi Ye, Zhangmingzu Qu, Xiaowu Yang, Chen Wang

**Affiliations:** 1Xi’an Key Laboratory of Textile Chemical Engineering Auxiliaries, School of Environmental and Chemical Engineering, Xi’an Polytechnic University, Xi’an 710048, China; 20190607@xpu.edu.cn (L.Q.); 41804010213@stu.xpu.edu.cn (Y.Z.); 42004010112@stu.xpu.edu.cn (Z.Q.); 2Key Laboratory of Auxiliary Chemistry and Technology for Chemical Industry, Ministry of Education, Shaanxi Collaborative Innovation Center of Industrial Auxiliary Chemistry & Technology, Shaanxi University of Science and Technology, Xi’an 710021, China; yangxiaowu@sust.edu.cn; 3Chemical Corporation of Changqing, Xi’an 710021, China; xuelilongcq@Petro-China.com.cn (X.L.); yez0_cq@Petro-China.com.cn (Z.Y.)

**Keywords:** morphological change, deteriorative monitoring, SEM investigation, site-directed reaction, sizing treatment

## Abstract

The O/W isocyanate emulsion can be used as a sizing agent to improve the waterproof performance of paper. However, the -NCO content in the emulsion diminishes with the prolongation of standing time. What is happening to this seemingly stable emulsion, especially concerning its microstructure evolution? We propose to monitor the emulsions deteriorative process by combining freeze-drying technique and SEM. Thus, the emulsion containing -NCO active group was obtained by the synthetic polymer emulsification of HDI trimers. The results of SEM demonstrate that the emulsion deteriorative process actually represents the collapsing and fusion of stable honeycomb structure with the prolongation of standing time and increasing temperature. This is possibly due to the fact that the inner aggregative HDI trimers are reacting with outside water to form urethane macromolecules, and this results in the collapsing and fusion of the honeycomb structure, as observed in SEM images. Moreover, the measurement results of -NCO content and FT-IR spectroscopy present the -NCO content as reducing with increasing standing time and temperature. This conclusion further proves our hypotheses. Additionally, the emulsions are used to treat the paper by site-directed reaction. The results show that the with the increase of the standing time and temperature, the contact angles and surface free energy show a decrease and an increase, respectively, whereas surface free energy appeared at a minimum of 29.19 mJ·m^−2^ when the standing time and temperature was 1 h and 25 °C.

## 1. Introduction

Hexamethylene diisocyanate (HDI) trimers emulsions fall into category of active emulsions since they should have free isocyanate (-NCO) contents in the emulsion to react with non-woven fabrics and paper [1,2,3]. Since this reaction could convert the hydrogen bonding into chemical bonds between fibers [4], paper can become a new high-strength, water-resistant, oil-resistant special paper after the treatment. This work can open up broader prospects for the application of paper in packaging, printing, construction, and other industrial and agricultural production [5,6].

The NCO groups can be so reactive that they react with almost all polar substances, so knowing the precise isocyanate content is important in aiding our evaluating of the quality of the materials. The reference test for the determination of NCO content is based on the modification of isocyanate functional groups to urea using dibuthyl amine solution in toluene followed by the titration of the excess dibuthyl amine with standardized hydrochloric acid solution. Other titration methods, such as using dicyclohexyl amine as an alternative to dibuthyl amine, have been developed [7,8,9]. However, there are a number of limitations. These methods are used for samples containing high isocyanate content. On the other hand, a large amount of sample is required when analyzing high molecular weight samples. Another limitation is that accurate titration cannot be performed in all solvents. Alternative methods for titration of simple isocyanates, such as using liquid chromatography, have been developed [10], but in many cases isocyanate the content of polymeric isocyanate cannot be determined by this method. FT-IR, middle infrared spectroscopy (MIR), and near infrared spectroscopy (NIR) has also been successfully utilized to determine NCO content of adhesive urethane prepolymers [11,12]. These methods have disadvantages such as poor accuracy of integration of peaks in infrared spectroscopy. Functional group analysis of polymers by NMR and ^19^F NMR spectroscopy is also known. Hydroxyl, amine, and acid functional groups of polymers have been successfully determined by ^19^F NMR spectroscopy [13,14,15]. The above-mentioned determination methods only emphasize on knowing the precise isocyanate content. The fact remains that we do not yet know the microstructure evolution caused by changes in isocyanate content. Therefore, to better understand the microstructure evolution process of emulsion with isocyanate content changes, we used freeze-drying and scanning electron microscopy to examine the microstructures evolution of isocyanate emulsion in different contents.

Given all this, the deteriorative process of the active emulsions is not still sufficiently studied and described in the literature. Therefore, the aim of the present work was to investigate the microstructure of active emulsions during their deteriorative processes with help of the combination of freeze-drying technique and SEM, which should be investigated in their original non-dried state [16] and allow us to obtain unique and valuable information about deteriorative process of the active emulsions.

## 2. Materials and Methods

### 2.1. Materials

Hexamethylene diisocyanate trimer, also named also known as 1,3,5-tris(6-iscoyanatohexyl)-1,3,5-triazine-2,4,6(1H,3H,5H)-trione, is a hexamethylene diisocyanate homopolymer. It was supplied by Kelude (Qingdao, China). 

Copolymer P(OEGMA-co-HEMA-co-BMA) was prepared by ATRP polymerization reactions. The synthesis and characterization of the copolymer were presented in literature [17] and the structure of the copolymer P(OEGMA-co-HEMA-co-BMA) is shown in Figure 1. The synthetic copolymer was used as an emulsifier to emulsify HDI trimer.

### 2.2. Preparation of Micrometer Emulsion 

The N-Methyl pyrrolidone dissolved copolymer P(OEGMA-co-HEMA-co-BMA) prepared can be used as a macro-emulsifier to emulsify HDI trimer. The 1.0 g copolymer P(OEGMA-co-HEMA-co-BMA) and 5.0 g HDI trimer were added in a 100 mL dried three-necked flask, which was stirred at 60 °C for 1 h to make reaction between -OH in P(OEGMA-co-HEMA-co-BMA) with -NCO in HDI trimer. When the temperature approached to 25 °C, 50.0 g water was added dropwise under stirring at 6000 r/min for 5 min. Then, a stable O/W emulsion was achieved. 

### 2.3. Characterization of Deteriorative Processes for Emulsion

#### 2.3.1. -NCO Content Measurement

The acetone-dibutylamine method was applied to test -NCO content. The sample (about 1.0000 g) was transferred to an erlenmeyer flask and was dissolved with 15 mL acetone. 10 mL acetone-dibutylamine solution was added to the above solution, which was stirred 5 min at 25 °C. Then, a few drops of bromocresol green indicator was added, and HCl standard solution was used to titrate. The titration end-point reached to termination when the color of solution changed from green to yellow. In order to guarantee the accuracy of experimental data, three samples in the same test environment were measured. The blank experiment is carried out in the meantime. The -NCO content was calculated according to Equation (1).
(1)$${\mathrm{wt}}_{\left(-CNO\right)}=(V_{0}-V_{1})\times C\times 4.202/m$$
where *C* was the molar concentration of standard NaOH and HCl solutions, *m* (g) was the weight of sample taken, and *V*_0_ and *V*_1_ were the volume of HCl used for the titrating the excessive dibutylamine of blank and sample, respectively.

#### 2.3.2. SEM Investigation

In order to monitor deteriorative processes of emulsion, the emulsions were drier using a freeze drier in liquid nitrogen in a vacuum environment. The dried emulsion samples were subjected to SEM using a scanning electron microscope (VEGA3, TESCAN, Brno, Czech Republic) to observe the microstructures of the emulsion. These samples were glued on an aluminum stub, and the surfaces were coated with gold.

#### 2.3.3. FT-IR Spectroscopy

The deteriorative processes of emulsion were also monitored by FT-IR spectroscopy. The dried emulsion samples were ground into power to measure. The FT-IR spectroscopy of powers were recorded employing a Fourier transform infrared spectrophotometer (Model VERTEX70, Bruker, Karlsruhe, Germany) in the range of 4000–500 cm^−1^ as a potassium bromide pellet technique. The monitor of deteriorative processes was realized by the ratio of the integral area of isocyanate group (-N=C=O) to the integral area of carbamate group (-NH-COO^−^).

### 2.4. Characterization of Sizing-Treated Paper Fabric

The untreated paper with a basis weight of about 80 g/m^2^ was formed by cotton pulp. It (5 × 5 cm^2^) was immersed in the sizing agent micrometer emulsion with a concentration of 50.0% for 10 min, and then pressed between squeezing rolls to remove the excess liquid to reach about 95% wet pickup. Subsequently, the sizing-treated paper was dried at 105 ℃ for 10 min. The blank sample was made through the same treating procedure, except for using the deionized water instead of the micrometer emulsion. By comparing the change in quality of sizing before and after, the mass ratio of sizing agent to paper was 2.0%.

#### 2.4.1. Dynamic Contact Angles

The dynamic contact angle between distilled water and the surface of the sizing-treated paper fabric was measured at room temperature by a FIBRO 1100 DAT dynamic contact angle and penetrability analyzer (DCA, Stockholm, Sweden). The volume of the distilled water was 4 mL and the resistance was 18.5 M. The breadth of the samples was 15 mm. 

#### 2.4.2. Surface Free Energy 

Contact angles are measured at 25 °C, and the results reported are the mean values of 5 replicates. The preferable equation to calculate the surface free energy, which is also surface free energy between polymers and an ordinary liquid, is as follows in Equation (2).
(2)$$\left(1+\cos\theta_{\mathrm{liquid}})\gamma_{\mathrm{liquid}}=4(\frac{\gamma_{\mathrm{liquid}}^{d}\gamma^{d}}{\gamma_{\mathrm{liquid}}^{d}+\gamma^{d}}+\frac{\gamma_{\mathrm{liquid}}^{p}\gamma^{p}}{\gamma_{\mathrm{liquid}}^{p}+\gamma^{p}}\right)$$
in which $\gamma =\gamma^{d}+\gamma^{p}$, $\gamma_{\mathrm{liquid}}=\gamma_{\mathrm{liquid}}^{d}+\gamma_{\mathrm{liquid}}^{p}$, γ is the surface free energy, $\gamma^{d}$ is the dispersion component, $\gamma^{p}$ is polar component, $\theta_{\mathrm{liquid}}$ is contact angle of the polymer with water or diiodomethane. The numerical values used are $\gamma_{H_{2}O}^{d}=22.1\;\mathrm{mJ}/m^{2}$, $\gamma_{H_{2}O}^{p}=50.7\;\mathrm{mJ}/m^{2}$, $\gamma_{CH_{2}I_{2}}^{d}=44.1\;\mathrm{mJ}/m^{2}$, $\gamma_{CH_{2}I_{2}}^{p}=6.70\;\mathrm{mJ}/m^{2}$.

## 3. Results and Discussion

### 3.1. Microstructure of Emulsion 

The copolymer P(OEGMA-co-HEMA-co-BMA) prepared can be used as an emulsifier to emulsify HDI trimers, which can be defined as a reactive macro-emulsifier. The -OH in HEMA can be reacted with HDI trimers and the water is added dropwise to achieve a stable O/W emulsion. What’s more, freeze-drying technique and SEM is combined to examine the exact microstructures of emulsion. Seen from Figure 2, the emulsion exhibits honeycomb structures. The reason about formation of this kind of structure is that the reactive macro-emulsifier divides the emulsion into two stable phases. One is the outer water phase with extended OEGMA segment and the other is inner aggregation areas of HDI trimers and organic solvents. As freeze-drying technique is used to remove the water, the lightweight polymer skeleton is formed. Moreover, the process of freeze-drying also removes the HDI trimers and organic solvents, which is the reason for the formation of the cellular honeycomb structures. The schematic diagram of the honeycomb structure is depicted in Figure 3.

### 3.2. Effect of Time on Deteriorative Processes of Emulsion

#### 3.2.1. -NCO Content and SEM Investigation

Detecting the -NCO content in the emulsions is a common method to investigate whether the emulsion undergoes deterioration. Seen from the digital photos in Figure 4, the appearance of emulsion does not change over time. However, the -NCO content is decreased obviously in the emulsion with the increase of storage time. It means that the emulsion has already loses its activity and becomes inactivated despite the unchanged appearance of the emulsion.

What is happened in the seemingly stable emulsion? SEM investigation is a novel and exact method to observe the deterioration of emulsion virtually. Micrographs of emulsions via different standing time are shown in Figure 5. The exact microstructure of emulsion with 1h standing time (Figure 5a) basically maintains the original honeycomb structures as shown in Figure 2, which indicates that the emulsion remains its activity. With increase of the time, the honeycomb structures are collapsed and the holes are fused. (Figure 5b–d). The possible reason is that copolymer P(OEGMA-co-HEMA-co-BMA) can be only a temporary physical skeleton to divided the emulsion into two phases and they will be partly fused due to molecular kinetic theory. Moreover, HDI trimers has active -NCO groups, which can be reacted with water as they are contacted. With the increase of standing time, more and more -NCO groups are reacted with outside water to form carbamates macromolecules and only the unreacted HDI trimers and organic solvents are removed to form random holes in the process of freeze-drying. Therefore, the honeycomb structures disappeared over standing time of emulsion and the emulsions with no honeycomb structures can be defined as inactivated emulsion or deteriorative emulsion. The deteriorative processes of emulsions can be interpreted in Figure 6.

#### 3.2.2. FT-IR Spectroscopy

Based on the discovery for the changing micrograph of emulsions, the FT-IR spectroscopy in Figure 7 and data in Table 1 confirms these results. Carbamate (-NH) groups and isocyanate (-N=C=O) groups appear in near 3500 and 2250 cm^−1^, respectively. Due to the reaction of HDI trimers with water, there will be more -NH groups and less -N=C=O groups in the system. With the increase of standing time, the peaks of -NH groups become wider and their values of integral area increase. To the contrary, the peaks of -N=C=O groups become smaller and narrower, and their values of integral area decrease appaerently. At this time, we difine the ratio of -NH groups integral area to -N=C=O groups integral area as S_(-NH)_:S_(-N=C=O)_. Seen from Table 1, the value of S_(-NH)_:S_(-N=C=O)_ increases from 1.219 to 5.207 when the standing time of emulsion changes from 1 h to 24 h. All the FT-IR data confims that more and more -NCO groups are reacted with outside water to form carbamates macromolecules with the increase of standing time of emulsion, which coincides with explanation of collapsing and fused of honeycomb structures. 

### 3.3. Effect of Temperature on Deteriorative Processes of Emulsion 

#### 3.3.1. -NCO Content and SEM Investigation 

The temperature is also an important factor for deterioration of emulsions. We heated the emulsion to different temperature for 10 min and restore the -NCO content at room temperature. When the temperature is from room temperature to 50 °C, the appearances of emulsions present stable state (digital photos in Figure 8), and the -NCO content decrease slightly (Figure 8). However, when the temperature is above 50 °C, the -NCO content decrease dramatically. As shown in the digital photos in Figure 8, the appearances of emulsions show separated phases when the temperature reaches to 70 °C and 90 °C. However, their appearances change to viscous and stable phases when the temperatures are cooling to 25 °C. Therefore, it is impossible to judge the degree of emulsion deterioration from the appearance alone when emulsion suffers from short time heat. 

Herein, SEM is used to investigate the exact structure of emulsions after suffering from short time heat. Figure 9 shows micrographs of emulsions suffering from 10 min heat via different temperature, which are tested using the samples at room temperature and corresponds to the digital photos in Figure 8. Although all of the appearances of testing samples are stable and no separation, the exact structure of the emulsion has already changed dramatically. At room temperature, the emulsion shows honeycomb structures (Figure 9a) which is depicted in Figure 3. Moreover, the stable honeycomb structures have done little damage when the emulsion suffers from 10 min heat with 50 °C (Figure 9b), which demonstrates that the emulsion do not deteriorate at this time. However, when the emulsion suffers from 10 min heat with 70 °C (Figure 9c), especially at 90 °C (Figure 9d), the honeycomb structures have collapsed and fused. The results are consistent with the data of existed -NCO content. The possible reason is that copolymer P(OEGMA-co-HEMA-co-BMA) can be only a temporary physical skeleton to divided the emulsion into two phases. With an increase of the treating temperature for emulsions, more and more -NCO groups is reacted with outside water to form carbamates macromolecules and the unreacted HDI trimers and organic solvents are removed to form random fused holes in the process of freeze-drying. Therefore, the honeycomb structures disappeared. Emulsions with strongly damaged honeycomb structures can be defined as inactivated emulsion or deteriorative emulsion. The deteriorative processes of emulsions can be interpreted in Figure 6. 

#### 3.3.2. FT-IR Spectroscopy

Like the analysis of FT-IR spectroscopy in Section 3.2.2, Table 2 shows integral areas of carbamate groups and isocyanate groups of emulsions in the FT-IR spectra, which suffer from 10 min heat via different temperature. Due to the reaction of HDI trimers with water, there will be more -NH groups and less -N=C=O groups in the system. With the increase of treating temperature, the values of integral areas (-NH groups) increase. On the contrary, the values of integral areas (-N=C=O groups) decrease correspondingly. The value of S_(-NH)_:S_(-N=C=O)_ increases from 1.219 to 2.715 and does not increase so much as when the treating temperature of emulsion increases from room temperature to 50 °C. Combined with results of SEM, the exact emulsion microstructures have largely kept the honeycomb structures, depicting that the emulsion should be highly active at this time. However, when the treating temperature of emulsion increaes from 50 °C to 70 °C, then to 90 °C, the value of S_(-NH)_:S_(-N=C=O)_ increases dramatically to 4.043 and 8.603, respectively, showing that more and more -NCO groups are reacted with outside water to form carbamates macromolecules. This coincides with explanation of collapsing and fused of honeycomb structures. 

### 3.4. The Site-Directed Reaction of Active Substances with Paper Fibers

As shown in Figure 10, the hydroxyl groups between the paper sheets are hydrogen-bonded, and the hydrogen bonding is not only low in bond energy but also easily destroyed by water molecules, showing that the wet strength of plain paper is relatively low. On the other hand, after the emulsion-treated paper, the hydroxyl groups between the fibers reacted with -NCO to convert hydrogen bonds into chemical bonds, which enhanced the strength of the paper and overcome the shortcomings of traditional paper strength is not high.

#### 3.4.1. Dynamic Contact Angles 

Compared with the blank sample, the contact angle of the emulsion significantly increased as shown in Figure 11. For example, the contact angle of a blank sample is quickly decreases from 92.5° (in 0.2 s) to 0° (in 4.8 s). When left at 25 °C for 1 h, the hydrophobic segments can form a hydrophobic film on the fiber surface and change the morphology of the fiber surface, so the sample shows excellent hydrophobicity. As the residence time and temperature of the emulsion increase, more and more -NCO groups react with external water to form urethane macromolecules, resulting in a decrease in the activity of the emulsion (compare with the SEM image of Figure 9). When the temperature rises to 70 °C and then drop to 25 °C, and the contact angle of the emulsion does not change. Since the reaction of -NCO groups with external water to form urethane macromolecules is an irreversible reaction, the emulsion after high temperature has been changed, so the contact angle does not increase after a significant decline.

#### 3.4.2. Surface Free Energy

Table 3 shows the surface free energy (calculated by the contact angle with water and diiodomethane) of sized paper fabrics of different emulsions. The coating surface has a higher *γ*^d^ and a smaller *γ*^p^, indicating that there is a large contact angle between the polar liquid and the non-polar coating film. At the same time, the surface free energy increased from 29.19 mJ·m^−2^ to 38.81 mJ·m^−2^. The surface free energy of the film is increased by 32.96%. The reason for these apparent changes may involve increasing the number of hydrophilic units in the copolymer [18]. The results show that as the temperature and time for the emulsion increase, more and more -NCO groups react with external water to form urethane macromolecules, which destroys the structure of the emulsion and decreases the water resistance.

## 4. Conclusions

The emulsion containing -NCO active group was obtained by the emulsification of Hexamethylene diisocyanate (HDI) trimers, and it can be used as a sizing agent to improve the waterproof performance of paper. However, the -NCO content is diminishing with the prolongation of standing time and increasing temperature. What is happened to this seemingly stable emulsion especially microstructure evolution? Herein, we combined freeze-drying technique and SEM to monitoring the emulsions’ deteriorative process. The results demonstrate that the deteriorative process is actually the collapsing and fusion of stable honeycomb structure with the prolongation of standing time and increasing temperature. This is possibly due to the fact that the inner aggregation HDI trimmers are reacted with outside water to form urethane macromolecules, and result in collapsing and fusion of honeycomb structure, as observed in SEM images. Moreover, the measurement results of -NCO content and FT-IR spectroscopy present the -NCO content is reducing with increasing standing time and temperature. This method of combining freeze-drying technique and SEM may be useful tool to monitor and study microstructure evolution and may be important in aiding our understanding of the mechanism of chemical processes. In addition, the stable emulsions are used to treat the paper by site-directed reaction. The results show that the with the increase of the standing time and temperature, the contact angles and surface free energy show a decrease and an increase, respectively, whereas surface free energy appeared a minimum 29.19 mJ·m^−2^ when the standing time and temperature was 1 h and 25 °C.

## Figures and Tables

**Figure 1 molecules-26-06471-f001:**
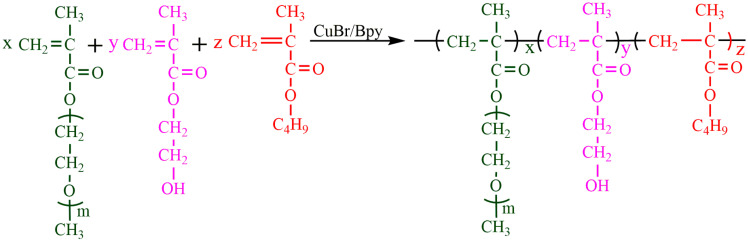
The structure of copolymer P(OEGMA-co-HEMA-co-BMA).

**Figure 2 molecules-26-06471-f002:**
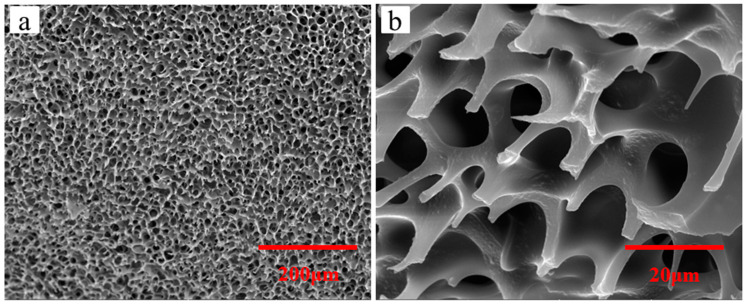
Micrograph of emulsion at (**a**) low- and (**b**) high-magnification.

**Figure 3 molecules-26-06471-f003:**
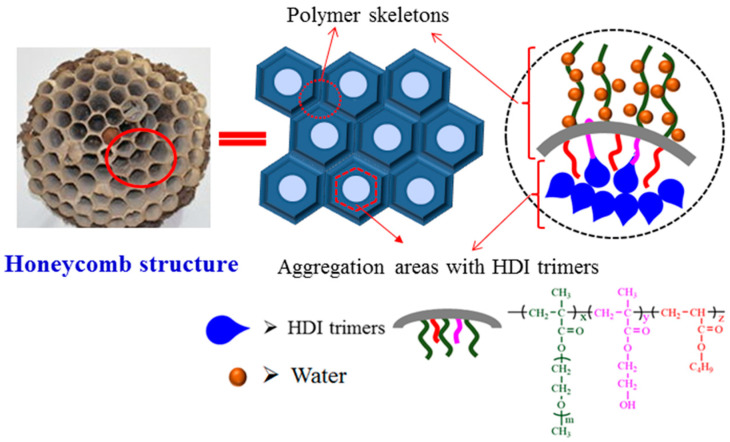
Schematic diagram of the honeycomb structure.

**Figure 4 molecules-26-06471-f004:**
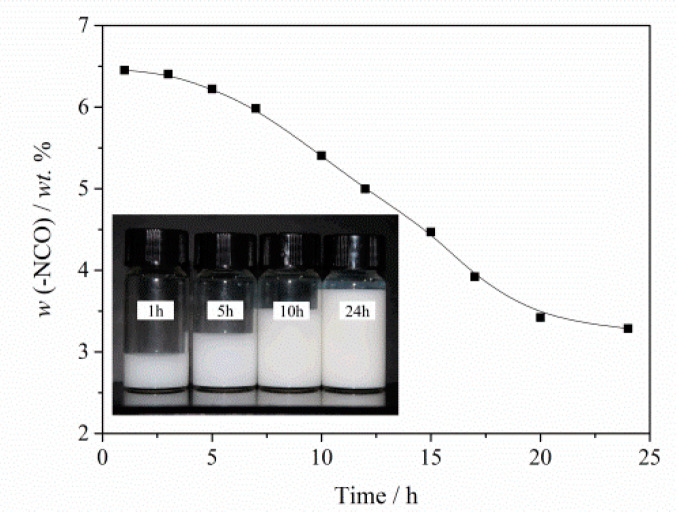
-NCO content in the emulsions via different standing time and their digital photos.

**Figure 5 molecules-26-06471-f005:**
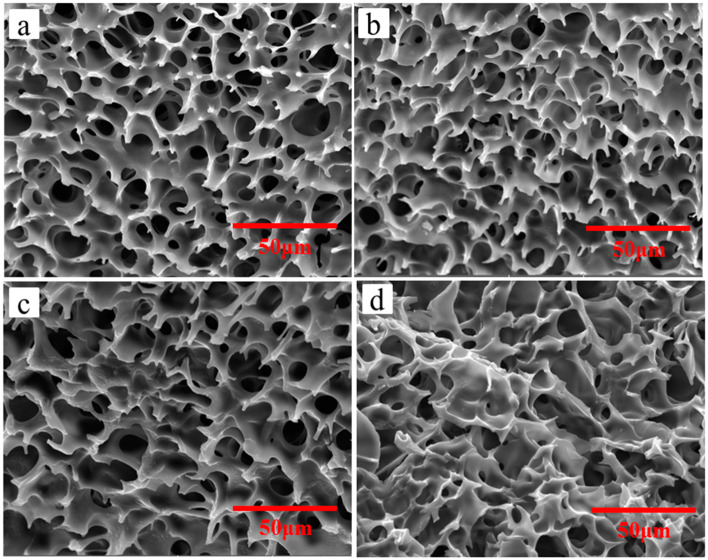
Micrograph of emulsions via different standing time: (**a**) 1 h; (**b**) 5 h; (**c**) 10 h; and (**d**) 24 h.

**Figure 6 molecules-26-06471-f006:**
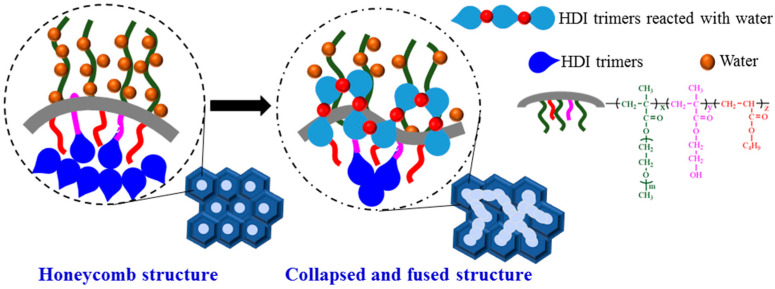
Schematic process of the collapsing and fusion of honeycomb structure.

**Figure 7 molecules-26-06471-f007:**
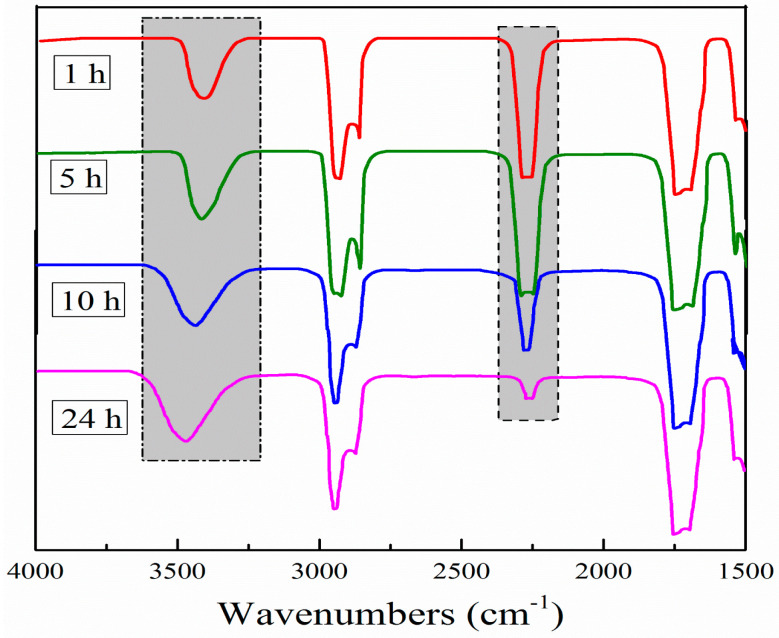
FT-IR spectra of emulsion with different standing time.

**Figure 8 molecules-26-06471-f008:**
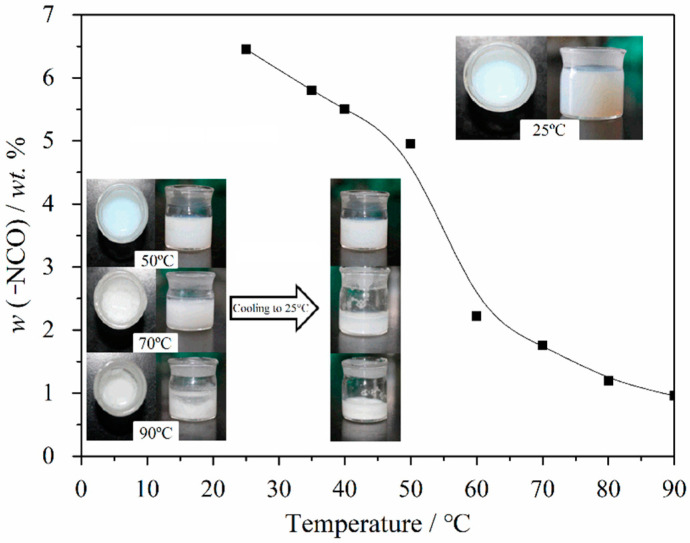
-NCO content in the emulsions suffering from 10 min heat via different temperature and their digital photos.

**Figure 9 molecules-26-06471-f009:**
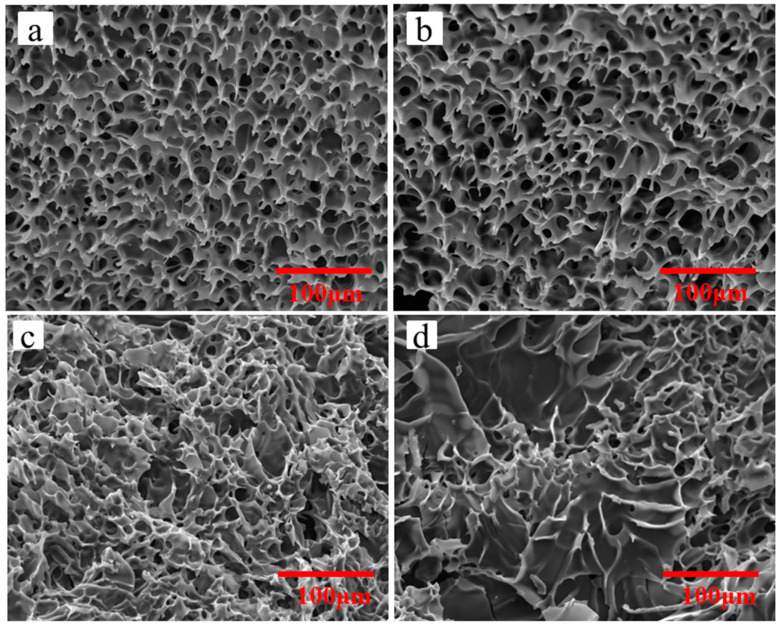
Micrograph of emulsions suffering from 10 min heat via different temperature: (**a**) 25 °C; (**b**) 50 °C; (**c**) 70 °C; and (**d**) 90 °C.

**Figure 10 molecules-26-06471-f010:**
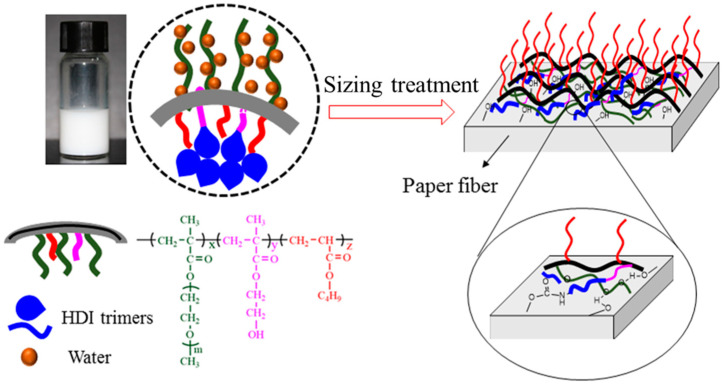
Schematic diagram of site-directed reaction of active material and paper fiber.

**Figure 11 molecules-26-06471-f011:**
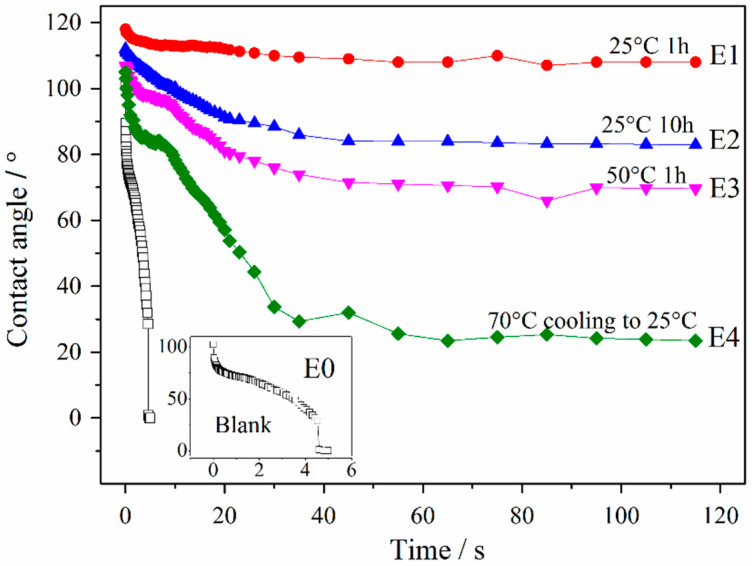
The dynamic contact angle of the sizing-treated paper fabric with different emulsions.

**Table 1 molecules-26-06471-t001:** Integral area of carbamate groups (-NH) and isocyanate groups (-N=C=O) of emulsions with different standing time in FT-IR spectra.

Samples	S_(-NH)_	S_(-N=C=O)_	S_(-NH)_:S_(-N=C=O)_
1 h	239.81	199.89	1.219
5 h	243.16	196.84	1.235
10 h	307.01	133.21	2.305
24 h	369.02	70.87	5.207

**Table 2 molecules-26-06471-t002:** Integral area of carbamate groups (-NH) and isocyanate groups (-N=C=O) of emulsions suffering from 10min heat via different temperature in FT-IR spectra.

Samples	S_(-NH)_	S_(-N=C=O)_	S_(-NH)_:S_(-N=C=O)_
25 °C	239.81	199.89	1.219
50 °C	315.23	116.68	2.715
70 °C	353.79	87.59	4.043
90 °C	394.63	45.87	8.603

**Table 3 molecules-26-06471-t003:** Surface free energy of the sizing-treated paper fabric with different emulsions.

Emulsion Sample	Contact Angles/°	Surface Free Energy/(mJ·m^−2^)
Water	Diiodomethane	*γ* ^d^	*γ* ^p^	*γ*
E1	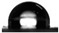 109.6	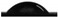 58.7	27.10	2.09	29.19
E2	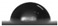 91.2	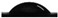 52.2	28.10	3.14	31.24
E3	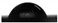 75.3	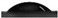 47.3	29.89	4.29	34.18
E4	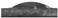 33.7	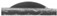 30.1	34.40	4.41	38.81

## Data Availability

The data presented in this study are available from the authors.

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
