# Peer review of "Scanning Electron Microscopy Investigation for Monitoring the Emulsion Deteriorative Process and Its Applications in Site-Directed Reaction with Paper Fabric"

_molecules, 2021, doi:10.3390/molecules26216471_

Round 1

Reviewer 1 Report

The work entitled ¨SEM Investigation for Deteriorative Monitoring of Emulsion and Its Applications in Site-Directed Reaction with Paper Fabric¨ is interesting. The authors used the freeze-drying technique and scanning electron microscopy (SEM) to monitor deteriorative processes of emulsions with the effects of time or temperature.

I feel that a comprehensive revision of the work is needed to be considered in the journal. This can be an inconvenience for the authors. However, please note that the work was not ready to be submitted to the journal. Note that abbreviations (SEM) should be avoided in the title. 

From abstract   <<and paper designed for.>>  Some readers may object to a preposition such as for at the end of a sentence. Consider rewording the sentence if your readers are likely to object.

<<of paper is larger and the surface free energy is lower>>, It appears that this sentence includes an incomplete comparison. Consider rewriting to complete the comparison.

The abstract should be rewritten to meet the following criteria. Indicate the gap in the knowledge that your work aims to fill. Then, indicate the main conclusion and the relationship with the aim of the work.

For consistency, the abbreviation (i.e., SEM) should not be repeated through the work, only the first time they are mentioned.

The introduction is short and poorly drafted. The discussion of the work must be improved. Also, the conclusions should be rewritten. The Conclusions section not only reviews the results or observations but it also interprets them. In this section, you can therefore point out the following: What is significant? Why the results or observations are valid?

The Figures are very well designed.

Please revise the article carefully. Be sure that all mistakes were corrected before submitting to this or another journal. 

Author Response

Dear Editors and Reviewers:

Thank you for your letter and for the comments concerning our manuscript “Scanning electron microscopy investigation for monitoring the emulsion deteriorative process and its applications in site-directed reaction with paper fabric(ID: molecules-1402168). Your comments were valuable and helpful for revising and improving the paper. We have considered them carefully and have made corrections that we hope will meet with your approval. The revised portions are marked in red in the paper. The main corrections in the paper, and our responses to the reviewers’ comments, are as follows:

Responds to Reviewer’s comments:

  1. Note that abbreviations (SEM) should be avoided in the title.

RESPONSE: Thank you for your feedback. We have changed the title into “Scanning electron microscopy investigation for monitoring the emulsion deteriorative process and its applications in site-directed reaction with paper fabric.

  1. From abstract <<and paper designed for.>> Some readers may object to a preposition such as for at the end of a sentence. Consider rewording the sentence if your readers are likely to object. <<of paper is larger and the surface free energy is lower>>, It appears that this sentence includes an incomplete comparison. Consider rewriting to complete the comparison.

The abstract should be rewritten to meet the following criteria. Indicate the gap in the knowledge that your work aims to fill. Then, indicate the main conclusion and the relationship with the aim of the work.

RESPONSE: Thank you for your suggestion. We have revised this sentence and rewritten the abstract, as follows.

The O/W emulsions of HDI trimers can be used as a sizing agent to improve the waterproof performance of paper. However, the –NCO content in emulsion is diminishing with the prolongation of standing time. What is happened to this seemingly stable emulsion, especially microstructure evolution? We propose to monitor emulsions deteriorative process by combining freeze-drying technique and SEM. So the emulsion containing –NCO active group was obtained by the synthetic polymer emulsification of HDI trimers. The results of SEM demonstrate that emulsion deteriorative process is actually the collapsing and fusion of stable honeycomb structure with the prolongation of standing time and increasing temperature. This possibly because of inner aggregative HDI trimers are reacted with outside water to form urethane macromolecules, and result in collapsing and fusion of honeycomb structure, as observed in SEM images. Moreover, the measurement results of –NCO content and FT-IR spectroscopy present the –NCO content is reducing with increasing standing time and temperature. It also can be used to confirm our deduction. Additionally, the emulsions are used to treat the paper by site-directed reaction. The results show that the with the increase of the standing time and temperature the contact angles and surface free energy show a decrease and an increase respectively, whereas surface free energy appeared a minimum 29.19mJ•m-2 when the standing time and temperature was 1h and 25℃.

  1. For consistency, the abbreviation (i.e., SEM) should not be repeated through the work, only the first time they are mentioned.

RESPONSE: Thank you for your feedback. We have only retained the first time mentioned. Abbreviation, and deleted all repeated abbreviation.

  1. The introduction is short and poorly drafted.

RESPONSE: Thank you for your suggestion. We have expanded the introduction section, as follows.

Hexamethylene diisocyanate (HDI) trimers emulsions fall into category of active emulsions because they should have free isocyanate (–NCO) contents in the emulsion to react with non-woven fabrics and paper [1-3]. Because this react could convert the hydrogen bonding into chemical bonds between fibers[4], paper can become a new high-strength, water-resistant, oil-resistant special paper after the treatment. This work can open up broader prospects for the application of paper in packaging, printing, construction and other industrial and agricultural production[5-6].

The NCO groups can be so reactive that they react with almost all polar substances, so knowing the precise isocyanate content is important in aiding our evaluating quality of the materials. The reference test for determinationof NCO content is based on the modification of isocyanate functional groups to urea using dibuthyl amine solution in toluene followed by the titration of the excess dibuthyl amine with standardized hydrochloric Acid solution. Other titration methods, such as using dicyclohexyl amine as an alternative to dibuthyl amine, have been developed[7-9]. However, there are a number of limitations. These methods are used for samples containing high isocyanate content. On the other hand, a large amount of sample is required when analyzing high molecular weight samples. Another limitation is that accurate titration cannot be performed in all solvents. Alternative methods for titration of simple isocyanates, such as using liquid chromatography, have been developed[10], but in many cases isocyanate content of polymeric isocyanate cannot be determined by this method. FT-IR, middle infrared spectroscopy (MIR) and near infrared spectroscopy (NIR) spectroscopy has also been successfully utilized to determine NCO content of adhesive urethane prepolymers [11–12]. These methods have disadvantages such as poor accuracy of integration of peaks in infrared spectroscopy. Functional group analysis of polymers by NMR and 19F NMR spectroscopy is also known. Hydroxyl, amine and acid functional groups of polymers have been successfully determined by 19F NMR spectroscopy [13–15].the above-mentioned determination methods only emphasis on knowing the precise isocyanate content, the fact is we don't know the microstructure evolution caused by content isocyanate change. Herein, to better understand the microstructure evolution process of emulsion with content isocyanate change, we used freeze-drying and scanning electron microscopy (SEM) to examine the microstructures evolution of isocyanate emulsion in different content.

Given all this, the deteriorative process of the active emulsions is not still sufficiently studied and described in the literature. Therefore, the aim of the present work was to investigate microstructure of active emulsions during their deteriorative processes with help of the combination of freeze-drying technique and SEM, which should be exactly investigated in original non-dried state [16] and allow us to obtain unique and valuable information about deteriorative process of the active emulsions.

  1. The conclusions should be rewritten. The Conclusions section not only reviews the results or observations but it also interprets them. In this section, you can therefore point out the following: What is significant? Why the results or observations are valid?

RESPONSE: Thank you for your feedback. We have revised the “Conclusions”, as follows.

The emulsion containing –NCO active group was obtained by the emulsification of Hexamethylene diisocyanate (HDI) trimers, and it can be used as a sizing agent to improve the waterproof performance of paper. However, the –NCO content is diminishing with the prolongation of standing time and increasing temperature. What is happened to this seemingly stable emulsion especially microstructure evolution? Herein, we combined freeze-drying technique and SEM to monitoring emulsions deteriorative process. The results demonstrate that deteriorative process is actually the collapsing and fusion of stable honeycomb structure with the prolongation of standing time and increasing temperature. This possibly because of inner aggregation HDI trimmers are reacted with outside water to form urethane macromolecules, and result in collapsing and fusion of honeycomb structure, as observed in SEM images. Moreover, the measurement results of –NCO content and FT-IR spectroscopy present the –NCO content is reducing with increasing standing time and temperature. This method of combining freeze-drying technique and SEM may be useful tool to monitor and study microstructure evolution and may be important in aiding our understanding of the mechanism of chemical processes. In addition, the stable emulsions are used to treat the paper by site-directed reaction. The results show that the with the increase of the standing time and temperature the contact angles and surface free energy show a decrease and an increase respectively, whereas surface free energy appeared a minimum 29.19mJ•m-2 when the standing time and temperature was 1h and 25℃.

We have tried our best to improve the manuscript. Moreover, we made corresponding corrections to the manuscript, as shown in red in the revised paper. We earnestly appreciate the Editors/Reviewers’ kind comments, and we hope that the corrections will meet with your approval. Once again, thank you very much for your comments and suggestions.

Thank you and best regards. 
Yours sincerely,
Liewei Qiu, Yongkang Zhang, Xueli Long, Zhi Ye, Zhangmingzu Qu, Xiaowu Yang, Chen Wang
Corresponding author:
Name: Chen Wang

Reviewer 2 Report

The paper entitled ‘SEM Investigation for Deteriorative Monitoring of Emulsion and Its Applications in Site-Directed Reaction with Paper Fabric’ needs major revision.

  1. Some small spelling mistakes appear in the text.
  2. Improve the description of the Introduction. Improve the description of the motivation of this work.
  3. Add more details of the HDI trimer.
  4. Which solvent has been used to dissolve P(OEGMA-co-HEMA-co-BMA) ?? Should be added.

Revise this sentence: ‘The solvent dissolved copolymer P(OEGMA-co-HEMA-co-BMA) prepared can be used as a macro-emulsifier to emulsify HDI trimer.’

  1. What was the mass of samples for the -NCO content measurement??
  2. Figure 4 – add standard deviation. How many samples were tested?
  3. Conclusion – should be rebuilt. Insert some results.

Author Response

Dear Editors and Reviewers:

Thank you for your letter and for the comments concerning our manuscript “Scanning electron microscopy investigation for monitoring the emulsion deteriorative process and its applications in site-directed reaction with paper fabric(ID: molecules-1402168). Your comments were valuable and helpful for revising and improving the paper. We have considered them carefully and have made corrections that we hope will meet with your approval. The revised portions are marked in red in the paper. The main corrections in the paper, and our responses to the reviewers’ comments, are as follows:

Responds to Reviewer’s comments:

  1. Some small spelling mistakes appear in the text.

RESPONSE: Thank you for your feedback. We have checked spelling and tried our best to improve the language throughout the entire manuscript.

  1. Improve the description of the Introduction. Improve the description of the motivation of this work.

RESPONSE: Thank you for your feedback. We have further expanded the introduction section, as follows.

Hexamethylene diisocyanate (HDI) trimers emulsions fall into category of active emulsions because they should have free isocyanate (–NCO) contents in the emulsion to react with non-woven fabrics and paper [1-3]. Because this react could convert the hydrogen bonding into chemical bonds between fibers[4], paper can become a new high-strength, water-resistant, oil-resistant special paper after the treatment. This work can open up broader prospects for the application of paper in packaging, printing, construction and other industrial and agricultural production[5-6].

The NCO groups can be so reactive that they react with almost all polar substances, so knowing the precise isocyanate content is important in aiding our evaluating quality of the materials. The reference test for determinationof NCO content is based on the modification of isocyanate functional groups to urea using dibuthyl amine solution in toluene followed by the titration of the excess dibuthyl amine with standardized hydrochloric Acid solution. Other titration methods, such as using dicyclohexyl amine as an alternative to dibuthyl amine, have been developed[7-9]. However, there are a number of limitations. These methods are used for samples containing high isocyanate content. On the other hand, a large amount of sample is required when analyzing high molecular weight samples. Another limitation is that accurate titration cannot be performed in all solvents. Alternative methods for titration of simple isocyanates, such as using liquid chromatography, have been developed[10], but in many cases isocyanate content of polymeric isocyanate cannot be determined by this method. FT-IR, middle infrared spectroscopy (MIR) and near infrared spectroscopy (NIR) spectroscopy has also been successfully utilized to determine NCO content of adhesive urethane prepolymers [11–12]. These methods have disadvantages such as poor accuracy of integration of peaks in infrared spectroscopy. Functional group analysis of polymers by NMR and 19F NMR spectroscopy is also known. Hydroxyl, amine and acid functional groups of polymers have been successfully determined by 19F NMR spectroscopy [13–15].the above-mentioned determination methods only emphasis on knowing the precise isocyanate content, the fact is we don't know the microstructure evolution caused by content isocyanate change. Herein, to better understand the microstructure evolution process of emulsion with content isocyanate change, we used freeze-drying and scanning electron microscopy (SEM) to examine the microstructures evolution of isocyanate emulsion in different content.

Given all this, the deteriorative process of the active emulsions is not still sufficiently studied and described in the literature. Therefore, the aim of the present work was to investigate microstructure of active emulsions during their deteriorative processes with help of the combination of freeze-drying technique and SEM, which should be exactly investigated in original non-dried state [16] and allow us to obtain unique and valuable information about deteriorative process of the active emulsions.

  1. Add more details of the HDI trimer.

RESPONSE: The hexamethylene diisocyanate (HDI) trimer also known as 1,3,5-tris(6-iscoyanatohexyl)-1,3,5-triazine-2,4,6(1H,3H,5H)-trione, is a hexamethylene diisocyanate homopolymer.

  1. Which solvent has been used to dissolve P(OEGMA-co-HEMA-co-BMA) ? Should be added.

Revise this sentence: ‘The solvent dissolved copolymer P(OEGMA-co-HEMA-co-BMA) prepared can be used as a macro-emulsifier to emulsify HDI trimer.’

RESPONSE: The N-Methyl pyrrolidone was used to dissolve P(OEGMA-co-HEMA-co-BMA). And This sentence: ‘The solvent dissolved copolymer P(OEGMA-co-HEMA-co-BMA) prepared can be used as a macro-emulsifier to emulsify HDI trimer.’, which has been changed to ‘The N-Methyl pyrrolidone dissolved copolymer P(OEGMA-co-HEMA-co-BMA) prepared can be used as a macro-emulsifier to emulsify HDI trimer.’.

  1. What was the mass of samples for the -NCO content measurement?

RESPONSE: The mass of samples is about 1.0000g.

  1. Figure 4 – add standard deviation. How many samples were tested?

RESPONSE: Figure 4 depicted the–NCO content in the emulsions via different standing time. In order to guarantee the accuracy of experimental data, three samples in the same test environment were measured.

  1. Conclusion – should be rebuilt. Insert some results.

RESPONSE: Thank you for your feedback. Some results have been inserted in the “Conclusion” as follows.

The emulsion containing –NCO active group was obtained by the emulsification of Hexamethylene diisocyanate (HDI) trimers, and it can be used as a sizing agent to improve the waterproof performance of paper. However, the –NCO content is diminishing with the prolongation of standing time and increasing temperature. What is happened to this seemingly stable emulsion especially microstructure evolution? Herein, we combined freeze-drying technique and SEM to monitoring emulsions deteriorative process. The results demonstrate that deteriorative process is actually the collapsing and fusion of stable honeycomb structure with the prolongation of standing time and increasing temperature. This possibly because of inner aggregation HDI trimmers are reacted with outside water to form urethane macromolecules, and result in collapsing and fusion of honeycomb structure, as observed in SEM images. Moreover, the measurement results of –NCO content and FT-IR spectroscopy present the –NCO content is reducing with increasing standing time and temperature. This method of combining freeze-drying technique and SEM may be useful tool to monitor and study microstructure evolution and may be important in aiding our understanding of the mechanism of chemical processes. In addition, the stable emulsions are used to treat the paper by site-directed reaction. The results show that the with the increase of the standing time and temperature the contact angles and surface free energy show a decrease and an increase respectively, whereas surface free energy appeared a minimum 29.19mJ•m-2 when the standing time and temperature was 1h and 25℃.

We have tried our best to improve the manuscript. Moreover, we made corresponding corrections to the manuscript, as shown in red in the revised paper. We earnestly appreciate the Editors/Reviewers’ kind comments, and we hope that the corrections will meet with your approval. Once again, thank you very much for your comments and suggestions.

Thank you and best regards. 
Yours sincerely,
Liewei Qiu, Yongkang Zhang, Xueli Long, Zhi Ye, Zhangmingzu Qu, Xiaowu Yang, Chen Wang
Corresponding author:
Name: Chen Wang

Reviewer 3 Report

In the evaluation of hydrophilicity/hydrophobicity of sizing agent-treated paper by dynamic contact angle measurement, quantitative information on the sizing agent incorporated into the paper fabric would be required.

Author Response

Dear Editors and Reviewers:

Thank you for your letter and for the comments concerning our manuscript “Scanning electron microscopy investigation for monitoring the emulsion deteriorative process and its applications in site-directed reaction with paper fabric(ID: molecules-1402168). Your comments were valuable and helpful for revising and improving the paper. We have considered them carefully and have made corrections that we hope will meet with your approval. The revised portions are marked in red in the paper. The main corrections in the paper, and our responses to the reviewers’ comments, are as follows:

Responds to Reviewer’s comments:

In the evaluation of hydrophilicity/hydrophobicity of sizing agent-treated paper by dynamic contact angle measurement, quantitative information on the sizing agent incorporated into the paper fabric would be required.

RESPONSE: Thank you for your feedback. The untreated paper with a basis weight of about 80 g/m2 was formed by cotton pulp. It (5×5 cm2) was immersed in the sizing agent micrometer emulsion with a concentration of 50.0% for 10 min, and then pressed between squeezing rolls to remove the excess liquid to reach about 95% wet pickup. Subsequently, the sizing-treated paper was dried at 105℃ for 10 min. The blank sample was made through the same treating procedure, except for using the deionized water instead of the micrometer emulsion. By comparing the change in quality of sizing before and after, the mass ratio of sizing agent to paper was 2.0%.

In addition, the above-mentioned part has been inserted into the manuscript.

We have tried our best to improve the manuscript. Moreover, we made corresponding corrections to the manuscript, as shown in red in the revised paper. We earnestly appreciate the Editors/Reviewers’ kind comments, and we hope that the corrections will meet with your approval. Once again, thank you very much for your comments and suggestions.

Thank you and best regards. 
Yours sincerely,
Liewei Qiu, Yongkang Zhang, Xueli Long, Zhi Ye, Zhangmingzu Qu, Xiaowu Yang, Chen Wang
Corresponding author:
Name: Chen Wang

Reviewer 4 Report

Qiu et al investigated the time- and temperature-dependent deterioration processes of active emulsions containing -NCO group using freeze-drying technique and scanning electron microscope. This work is pretty interesting and may attract the readership of Molecules. However, the current manuscript suffers from several major shortcomings which need to be addressed before it could be reconsidered for acceptance for publication.

  1. There is a spelling error in the title of the manuscript.
  2. The introduction section of the manuscript has not provided sufficient information on the current state of the field. The authors are suggested to further expand the introduction section.
  3. The novelty of the current work is not clear. The authors should further highlight the novelty of their work.
  4. The implication of the work is also not clear. The authors are suggested to discuss the implications of their findings with respect to the field in the conclusion section.
  5. The authors have not provided the number of measurements for all the experimental data (i.e., Figures 4, 8, 11, and Table 3). How many measurements did the authors carry out? The authors should perform the measurements two to three times and provide the mean ± standard deviation values.

Author Response

Dear Editors and Reviewers:

Thank you for your letter and for the comments concerning our manuscript “Scanning electron microscopy investigation for monitoring the emulsion deteriorative process and its applications in site-directed reaction with paper fabric(ID: molecules-1402168). Your comments were valuable and helpful for revising and improving the paper. We have considered them carefully and have made corrections that we hope will meet with your approval. The revised portions are marked in red in the paper. The main corrections in the paper, and our responses to the reviewers’ comments, are as follows:

Responds to Reviewer’s comments:

  1. There is a spelling error in the title of the manuscript.

RESPONSE: Thank you for your feedback. We have changed the title into “Scanning electron microscopy investigation for monitoring the emulsion deteriorative process and its applications in site-directed reaction with paper fabric. Meanwhile, we have checked spelling and revised in the manuscript.

  1. The introduction section of the manuscript has not provided sufficient information on the current state of the field. The authors are suggested to further expand the introduction section.

RESPONSE: Thank you for your suggestion. We have further expanded the introduction section, as follows.

Hexamethylene diisocyanate (HDI) trimers emulsions fall into category of active emulsions because they should have free isocyanate (–NCO) contents in the emulsion to react with non-woven fabrics and paper [1-3]. Because this react could convert the hydrogen bonding into chemical bonds between fibers[4], paper can become a new high-strength, water-resistant, oil-resistant special paper after the treatment. This work can open up broader prospects for the application of paper in packaging, printing, construction and other industrial and agricultural production[5-6].

The NCO groups can be so reactive that they react with almost all polar substances, so knowing the precise isocyanate content is important in aiding our evaluating quality of the materials. The reference test for determinationof NCO content is based on the modification of isocyanate functional groups to urea using dibuthyl amine solution in toluene followed by the titration of the excess dibuthyl amine with standardized hydrochloric Acid solution. Other titration methods, such as using dicyclohexyl amine as an alternative to dibuthyl amine, have been developed[7-9]. However, there are a number of limitations. These methods are used for samples containing high isocyanate content. On the other hand, a large amount of sample is required when analyzing high molecular weight samples. Another limitation is that accurate titration cannot be performed in all solvents. Alternative methods for titration of simple isocyanates, such as using liquid chromatography, have been developed[10], but in many cases isocyanate content of polymeric isocyanate cannot be determined by this method. FT-IR, middle infrared spectroscopy (MIR) and near infrared spectroscopy (NIR) spectroscopy has also been successfully utilized to determine NCO content of adhesive urethane prepolymers [11–12]. These methods have disadvantages such as poor accuracy of integration of peaks in infrared spectroscopy. Functional group analysis of polymers by NMR and 19F NMR spectroscopy is also known. Hydroxyl, amine and acid functional groups of polymers have been successfully determined by 19F NMR spectroscopy [13–15].the above-mentioned determination methods only emphasis on knowing the precise isocyanate content, the fact is we don't know the microstructure evolution caused by content isocyanate change. Herein, to better understand the microstructure evolution process of emulsion with content isocyanate change, we used freeze-drying and scanning electron microscopy (SEM) to examine the microstructures evolution of isocyanate emulsion in different content.

Given all this, the deteriorative process of the active emulsions is not still sufficiently studied and described in the literature. Therefore, the aim of the present work was to investigate microstructure of active emulsions during their deteriorative processes with help of the combination of freeze-drying technique and SEM, which should be exactly investigated in original non-dried state [16] and allow us to obtain unique and valuable information about deteriorative process of the active emulsions.

  1. The novelty of the current work is not clear. The authors should further highlight the novelty of their work.

RESPONSE: Currently, determination methods of isocyanate content only emphasis on knowing the precise isocyanate content, the fact is we don't know the microstructure evolution caused by content isocyanate change. Herein, to better understand the microstructure evolution process of emulsion with content isocyanate change, we used freeze-drying and scanning electron microscopy to examine the microstructures evolution of isocyanate emulsion in different content. Therefore, the aim of the present work was to investigate microstructure of active emulsions during their deteriorative processes with help of the combination of freeze-drying technique and SEM, which should be exactly investigated in original non-dried state and allow us to obtain unique and valuable information about deteriorative process of the active emulsions.

  1. The implication of the work is also not clear. The authors are suggested to discuss the implications of their findings with respect to the field in the conclusion section.

RESPONSE: Thank you for your suggestion. The implications of the work have been included in the conclusion section, as follows.

The emulsion containing –NCO active group was obtained by the emulsification of Hexamethylene diisocyanate (HDI) trimers, and it can be used as a sizing agent to improve the waterproof performance of paper. However, the –NCO content is diminishing with the prolongation of standing time and increasing temperature. What is happened to this seemingly stable emulsion especially microstructure evolution? Herein, we combined freeze-drying technique and SEM to monitoring emulsions deteriorative process. The results demonstrate that deteriorative process is actually the collapsing and fusion of stable honeycomb structure with the prolongation of standing time and increasing temperature. This possibly because of inner aggregation HDI trimmers are reacted with outside water to form urethane macromolecules, and result in collapsing and fusion of honeycomb structure, as observed in SEM images. Moreover, the measurement results of –NCO content and FT-IR spectroscopy present the –NCO content is reducing with increasing standing time and temperature. This method of combining freeze-drying technique and SEM may be useful tool to monitor and study microstructure evolution and may be important in aiding our understanding of the mechanism of chemical processes. In addition, the stable emulsions are used to treat the paper by site-directed reaction. The results show that the with the increase of the standing time and temperature the contact angles and surface free energy show a decrease and an increase respectively, whereas surface free energy appeared a minimum 29.19mJ•m-2 when the standing time and temperature was 1h and 25℃.

  1. The authors have not provided the number of measurements for all the experimental data (i.e., Figures 4, 8, 11, and Table 3). How many measurements did the authors carry out? The authors should perform the measurements two to three times and provide the mean ± standard deviation values.

RESPONSE: In order to guarantee the accuracy of experimental data, we have carried out three times to measure the –NCO content. And the experimental data of Figure 4and Figure 8 was the mean values. We have provided the mean values and standard deviation values as follows.

Table 1 The experimental data of–NCO content in the emulsions via different standing time

Time (h)

w (-NCO) /%

Mean values

Standard deviation values

1

2

3

1

6.40

6.55

6.40

6.45

0.07

3

6.35

6.52

6.33

6.40

0.09

5

6.20

6.35

6.11

6.22

0.10

7

5.90

6.11

5.93

5.98

0.09

10

5.42

5.50

5.28

5.40

0.09

12

4.93

5.13

4.94

5.00

0.09

15

4.42

4.61

4.38

4.47

0.10

17

3.84

4.03

3.89

3.92

0.08

20

3.32

3.52

3.42

3.42

0.08

24

3.19

3.40

3.28

3.29

0.09

Table 2 The experimental data of–NCO content in the emulsions via different temperature

Temperature (℃)

w (-NCO) /%

Mean values

Standard deviation values

1

2

3

25

6.36

6.56

6.43

6.45

0.08

30

6.11

6.32

6.20

6.21

0.09

35

5.69

5.92

5.79

5.80

0.09

40

5.44

5.69

5.49

5.54

0.11

45

4.99

5.18

5.10

5.09

0.08

50

4.51

4.57

4.60

4.56

0.04

The reported results of contact angles are the mean values of 5 replicates, and mean values of contact angles can be used to calculate surface free energy via preferable equation. So we have provided the mean values and standard deviation values of contact angles.

Table 3. The contact angles of polymer with water

Emulsion Sample

Contact Angles /º

Mean values

Standard deviation values

1

2

3

4

5

E1

108.2

109.9

110.4

110.9

108.6

109.6

1.04

E2

89.3

91.8

91.8

92.3

90.8

91.2

1.07

E3

74.1

75.2

75.9

76.8

74.5

75.3

0.97

E4

32.5

34.3

34.2

34.8

32.7

33.7

0.92

Table 3. The contact angles of polymer with diiodomethane

Emulsion Sample

Contact Angles /º

Mean values

Standard deviation values

1

2

3

4

5

E1

57.5

58.2

59.9

59.2

58.7

58.7

0.82

E2

50.9

51.8

52.7

53.2

52.4

52.2

0.79

E3

46.1

46.9

48.8

48.2

46.5

47.3

1.03

E4

29.2

29.8

31.2

30.9

29.4

30.1

0.80

We have tried our best to improve the manuscript. Moreover, we made corresponding corrections to the manuscript, as shown in red in the revised paper. We earnestly appreciate the Editors/Reviewers’ kind comments, and we hope that the corrections will meet with your approval. Once again, thank you very much for your comments and suggestions.

Thank you and best regards. 
Yours sincerely,
Liewei Qiu, Yongkang Zhang, Xueli Long, Zhi Ye, Zhangmingzu Qu, Xiaowu Yang, Chen Wang
Corresponding author:
Name: Chen Wang

Round 2

Reviewer 1 Report

The work was improved. However, still need a comprehensive review of the text. Please cite the abbreviation at the first appearance only. Also check spelling and editing errors.  1.0000 g is 1g. 

Author Response

Dear Reviewers:

Thank you for your letter and for the comments concerning our manuscript “Scanning electron microscopy investigation for monitoring the emulsion deteriorative process and its applications in site-directed reaction with paper fabric(ID: molecules-1402168). Your comments were valuable and helpful for revising and improving the paper. We have considered them carefully and have made corrections that we hope will meet with your approval. The revised portions are marked in red in the paper. The main corrections in the paper, and our responses to the reviewers’ comments, are as follows:

The work still needs a comprehensive review of the text. Please cite the abbreviation at the first appearance only. Also check spelling and editing errors. 1.0000 g is 1g.

RESPONSE: Thank you for your feedback. We have checked spelling and asked help from native speakers to revise the entire manuscript. Meanwhile, we have only retained the first time mentioned and deleted all repeated abbreviation.

The zeros after the decimal point in 1.0000g represent measurement precision.

We have tried our best to improve the manuscript. Moreover, we made corresponding corrections to the manuscript, as shown in red in the revised paper. We earnestly appreciate the Editors/Reviewers’ kind comments, and we hope that the corrections will meet with your approval. Once again, thank you very much for your comments and suggestions.

Thank you and best regards.
Yours sincerely,
Liewei Qiu, Yongkang Zhang, Xueli Long, Zhi Ye, Zhangmingzu Qu, Xiaowu Yang, Chen Wang
Corresponding author:
Name: Chen Wang

Reviewer 2 Report

Authors of this paper took into account all my comments and suggestion. Manuscript at the corrected version needs minor revision.

  1. In this work still appeared some editorial errors and spelling mistakes:

“This work can open up broader prospects for the application of paper in packaging, printing, construction and other industrial and agricultural production[5-6].”

“The NCO groups can be so reactive that they react with almost all polar substances, so knowing the precise isocyanate content is important in aiding our evaluating quality of the materials. The reference test for the determination of NCO content is based on the modification of isocyanate functional groups to urea using dibuthyl amine solution in toluene followed by the titration of the excess dibuthyl amine with standardized hydrochloric aAcid solution. Other titration methods, such as using dicyclohexyl amine as an alternative to dibuthyl amine, have been developed[7-9]. However, there are a number of limitations. These methods are used for samples containing high isocyanate content. On the other hand, a large amount of samples is required when analyzing high molecular weight samples. Another limitation is that accurate titration cannot be performed in all solvents. Alternative methods for titration of simple isocyanates, such as using liquid chromatography, have been developed[10], but in many cases, the isocyanate content of polymeric isocyanate cannot be determined by this method. FT-IR, middle infrared spectroscopy (MIR), and near infrared spectroscopy (NIR) spectroscopy has also been successfully utilized to determine NCO content of adhesive urethane prepolymers [11–12]. These methods have disadvantages such as poor accuracy of integration of peaks in infrared spectroscopy. Functional group analysis of polymers by NMR and 19F NMR spectroscopy is also known. Hydroxyl, amine, and acid functional groups of polymers have been successfully determined by 19F NMR spectroscopy [13–15].The above-mentioned determination methods only emphasis on knowing the precise isocyanate content, the fact is we don't know the microstructure evolution caused by content isocyanate change. Herein, to better understand the microstructure evolution process of emulsion with content isocyanate change, we used freeze-drying and scanning electron microscopy to examine the microstructures evolution of isocyanate emulsion in different content.”

  1. Other comments:

In the case of my comments about more details of the HDI trimer I meant e.g NCO content

Author Response

Dear  Reviewers:

Thank you for your letter and for the comments concerning our manuscript “Scanning electron microscopy investigation for monitoring the emulsion deteriorative process and its applications in site-directed reaction with paper fabric(ID: molecules-1402168). Your comments were valuable and helpful for revising and improving the paper. We have considered them carefully and have made corrections that we hope will meet with your approval. The revised portions are marked in red in the paper. The main corrections in the paper, and our responses to the reviewers’ comments, are as follows:

  1. In this work still appeared some editorial errors and spelling mistakes:

RESPONSE: Thank you for your feedback. We have checked spelling and asked help from native speakers to revise the entire manuscript.

  1. More details of the HDI trimer I meant e.g NCO content

RESPONSE: I'm sorry to misunderstand your meaning. HDI trimers was supplied by Kelude (Qingdao, China).The -NCO content was (21.8±0.3)% and the residual HDI monomers content was below 0.15%.

We have tried our best to improve the manuscript. Moreover, we made corresponding corrections to the manuscript, as shown in red in the revised paper. We earnestly appreciate the Editors/Reviewers’ kind comments, and we hope that the corrections will meet with your approval. Once again, thank you very much for your comments and suggestions.

Thank you and best regards.
Yours sincerely,
Liewei Qiu, Yongkang Zhang, Xueli Long, Zhi Ye, Zhangmingzu Qu, Xiaowu Yang, Chen Wang
Corresponding author:
Name: Chen Wang

Reviewer 4 Report

The authors have addressed most of my comments, except comment no. 5. Have the new tables 1, 2, and 3 been incorporated into the revised manuscript? It seems to me that the current manuscript still keeps the old version of the tables.

Author Response

Dear Reviewers:

Thank you for your letter and for the comments concerning our manuscript “Scanning electron microscopy investigation for monitoring the emulsion deteriorative process and its applications in site-directed reaction with paper fabric(ID: molecules-1402168). Your comments were valuable and helpful for revising and improving the paper. We have considered them carefully and have made corrections that we hope will meet with your approval. The revised portions are marked in red in the paper. The main corrections in the paper, and our responses to the reviewers’ comments, are as follows:

The authors have addressed most of my comments, except comment no. 5. Have the new tables 1, 2, and 3 been incorporated into the revised manuscript? It seems to me that the current manuscript still keeps the old version of the tables.

RESPONSE: Thank you for your suggestion. We have incorporated the data of new tables 1, 2, and 3 into the revised manuscript.

We have tried our best to improve the manuscript. Moreover, we made corresponding corrections to the manuscript, as shown in red in the revised paper. We earnestly appreciate the Editors/Reviewers’ kind comments, and we hope that the corrections will meet with your approval. Once again, thank you very much for your comments and suggestions.

Thank you and best regards.
Yours sincerely,
Liewei Qiu, Yongkang Zhang, Xueli Long, Zhi Ye, Zhangmingzu Qu, Xiaowu Yang, Chen Wang
Corresponding author:
Name: Chen Wang